# Assessing the ecological resilience of Ebola virus in Africa and potential influencing factors based on a synthesized model

Li Shen[1☉], Jiawei Song[1☉], Yibo Zhou[1☉], Xiaojie Yuan[2], Samuel Seery[3], Ting Fu[2], Xihao Liu[2], Yihong Liu[4], Zhongjun Shao[2]*, Rui Li[2]*, Kun Liu[2]*

1 School of Remote Sensing and Information Engineering, Wuhan University, Wuhan, China, 2 Department of Epidemiology, The Ministry of Education Key Lab of Hazard Assessment and Control in Special Operational Environment, The Shaanxi Provincial Key Laboratory of Environmental Health Hazard Assessment and Protection, The Shaanxi Provincial Key Laboratory of Free Radical Biology and Medicine, School of Public Health, The Fourth Military Medical University, Xi'an, China, 3 Faculty of Health and Medicine, Division of Health Research, Lancaster University, Lancaster, United Kingdom, 4 The Third Regiment, School of Basic Medicine, Fourth Military Medical University, Xi'an, China

☉ These authors contributed equally to this work.
* 13759981783@163.com (ZS); lirui58@mail2.sysu.edu.cn (RL); liukun5959@qq.com (KL)

## Abstract

### Background

The Ebola epidemic has persisted in Africa since it was firstly identified in 1976. However, few studies have focused on spatiotemporally assessing the ecological adaptability of this virus and the influence of multiple factors on outbreaks. This study quantitatively explores the ecological adaptability of Ebola virus and its response to different potential natural and anthropogenic factors from a spatiotemporal perspective.

### Methodology

Based on historical Ebola cases and relevant environmental factors collected from 2014 to 2022 in Africa, the spatiotemporal distribution of Ebola adaptability is characterized by integrating four distinct ecological models into one synthesized spatiotemporal framework. Maxent and Generalized Additive Models were applied to further reveal the potential responses of the Ebola virus niche to its ever-changing environments.

### Findings

Ebola habitats appear to aggregate across the sub-Saharan region and in north Zambia and Angola, covering approximately 16% of the African continent. Countries presently unaffected by Ebola but at increasing risk include Ethiopia, Tanzania, Côte d'Ivoire, Ghana, Cameroon, and Rwanda. In addition, among the thirteen key influencing factors, temperature seasonality and population density were identified as significantly influencing the ecological adaptability of Ebola. Specifically, those regions were prone to minimal seasonal variations in temperature. Both the potential anthropogenic influence and vegetation coverage demonstrate a rise-to-decline impact on the outbreaks of Ebola virus across Africa.

**Data availability statement:** The authors confirm that all data underlying the findings are fully available without restriction. All relevant data are within the paper and its Supporting Information files. Publicly available data (CC BY 4.0) has been uploaded to the Researchgate (https://www.researchgate.net/profile/Jiawei-Song-14/publication/386178115_Assessing_the_ecological_resilience_of_Ebola_virus_in_Africa_and_potential_influencing_factors_based_on_a_synthesized_model/data/6747e1be790d154bf9afacef/DataforManuscriptPLOSntd.zip?_tp=eyJjb-250ZXh0Ijp7ImZpcnN0UGFnZSI6InB1YmxpY-2F0aW9uIiwicGFnZSI6InB1YmxpY2F0aW9uli-wicG9zaXRpb24iOiJwYWdlIn19).

**Funding:** This work was partly supported by grants from the National Natural Science Foundation of China [grant number 42201448 to Li Shen and 82273689 to Zhongjun Shao], Fundamental Research Funds for the Central Universities [grant number 2042024kf0023 to Li Shen], Education Reform Project of Wuhan University and Three Aspects of Education Reform Projects in School of Remote Sensing and Information Engineering [grant number YGJY202305 to Li Shen]. The funders had no role in study design, data collection and analysis, decision to publish, or preparation of the manuscript.

**Competing interests:** The authors have declared that no competing interests exist.

## Conclusions

Our findings suggest new ways to effectively respond to potential Ebola outbreaks in Sub-Saharan Africa. We believe that this integrated modeling approach and response analysis provide a framework that can be extended to predict risk of other worldwide diseases from a similar epidemic study perspective.

### Author summary

As one of the most lethal viral neglected tropical diseases in human history, Ebola virus disease typically causes organ damage, high fever, internal bleeding, and diarrhea with a high transmission rate and a case fatality rate between 50 and 90%. Traditional mathematical models neglect to incorporate the spatial variation of Ebola pathogens to reveal its potential outbreak source and sink areas. In addition, previous studies mainly regard EVD as an infectious disease to predict its distribution change, but rarely consider specific transmission process of the virus, resulting in sudden and rapid predictions with relatively low accuracy. In this study, we conducted a comprehensive analysis to spatiotemporally investigate the ecological adaptability of Ebola virus and its response to different environmental factors by proposing an integrated model. We found that Ebola habitats show significant differences in regional distribution. Temperature seasonality and population density might significantly influence the ecological adaptability of Ebola. Our findings can provide scientific support for practical prevention and control of future Ebola outbreaks in potential risk area.

## 1. Introduction

Ebola virus disease (EVD) is an acute infectious disease caused by the Ebola virus, a member of the Filoviridae family [1,2]. It is one of the most lethal viral diseases, characterized by high fever, organ damage, internal bleeding, and a case fatality rate of 50-90% worldwide [3]. EVD was firstly discovered in Sudan and the Democratic Republic of Congo in Africa in 1976 [4], and has since emerged in Central and West African nations [5], affecting Uganda, Democratic Republic of Congo (DRC), and Nigeria [6]. However, over the past decade, three major pandemics have occurred, including the 2014 Ebola outbreak in West Africa, the 2018 outbreak in DRC, and the 2022 outbreak in Uganda, with the first one recorded as the largest outbreak to date [7,8].

Previous scholars have been dedicated to researching virus spread through interactions between the susceptible agents and infected animals or humans. A variety of models have been developed or improved by continuously optimizing parameters [9]. For example, ecological niche models, incorporating environmental factors as covariates and using presence-only data, can be used to identify diverse habitats suitable for the spillover of different ebolaviruses [10]. From a zoonotic perspective, the Susceptible-Infectious-Recovered (SIR) compartmental model has also been deployed to explore the dynamics of Ebola transmission between species and simulate Ebola trends [11–14]. Also, the Geographic Information System-based multi-criteria evaluation (GIS-MCE) model can generates maps of risk area for Ebola spillover by taking account the spatial and temporal variability [15]. In addition, epidemiological compartmental models are used to forecast the spatial and temporal risks associated with Ebola Virus Disease [16]. And time-series regression models have significant potential to extract the filovirus distribution for improving the overall accuracy of traditional models [17–19].

However, the majority of conventional mathematical models paid little attention to incorporate the spatial variability of Ebola pathogens, limiting their ability to accurately identify potential outbreak source and sink areas. Especially, reliance on a single model may compromise both the robustness and accuracy of the results, as well as result in a loss of information, due to the model's inherent limitations [20]. Besides, the conventional models have paid little attention to simultaneously examining the impacts of both natural and anthropogenic factors on the host movement from a spatiotemporal perspective. Moreover, most relevant studies fail to use a comprehensive spatiotemporal dataset on potential influencing factors and the latest case data, such as from the 2022 Uganda outbreak based on a unified spatiotemporal scale.

Therefore, this study is purposed to propose a synthesized spatiotemporal framework to quantitatively explore the dynamic distribution of Ebola outbreaks in Africa, as well as the driving impact from different factors. A comprehensive and robust analysis can be provided by strategically integrating four distinct ecological models and incorporating an extensive dataset of multi-facet environmental factors and the most current case information. There were three specific objectives: 1) to extract the spatiotemporal distribution of Ebola by establishing a synthesized prediction model; 2) to investigating the pivotal factors that can affect the suitable niches of Ebola virus; and 3) to reveal the ecological adaptability of Ebola virus by characterizing its response to the influencing factors.

## 2. Methods

### 2.1. Study area and status

Since the first recorded outbreaks in Sudan and the DRC in 1976, the transmission of the EVD has predominantly been observed within the African continent. This study focused on Western and Central African nations where the most recent outbreaks occurred. According to the World Health Organization (WHO), two major outbreaks have been recorded since 2014. The first occurred in West Africa with over 28,600 infections and 11,325 fatalities in 2014 [7,21]. The second was in 2018 in the DRC, and had a total of 3,481 cases with 2299 fatalities [22]. Since then, there have been sporadic, relatively small outbreaks across Africa, in the DRC from 2018 to 2022, Guinea in February 2021, and Uganda in September 2022 as shown in Fig 1. Due to the sporadic nature of Ebola outbreaks prior to 2014 [23], we focused our investigation on the ecological adaptability of the Ebola virus based on EVD outbreaks occurring after the 2014 West Africa epidemic.

### 2.2. Data preparation

**2.2.1. Surveyed EVD cases.** Long-term EVD surveillance data was used in this study and included cases from the 2014-2016 outbreaks (https://data.humdata.org/dataset/evd-cases-by-district), the 2018 DRC outbreak (https://data.humdata.org/dataset/ebola-cases-and-deaths-drc-north-kivu), and the 2022 outbreak in Uganda (https://global.health/). The first two groups of data were obtained from The Humanitarian Data Exchange (HDX) website, which serves as an open platform for facilitating easily accessible and useable data for this analysis. Epidemic data were primarily based on information provided by the WHO, along with real-time reporting by national health authorities of different countries. Data from the 2022 Uganda outbreak were acquired from the Global Health website, which is purported to provide accurate real-time disease data in the early stages of an outbreak, tracking cases for the first 100 days. Geographical points of each case were collected from the aforementioned sources and then abstracted by locating them in the center of a specific region or in proximity to the medical facilities. This pre-process helped to avoid potential conflicts or errors without affecting the reliability and accuracy of the final results.

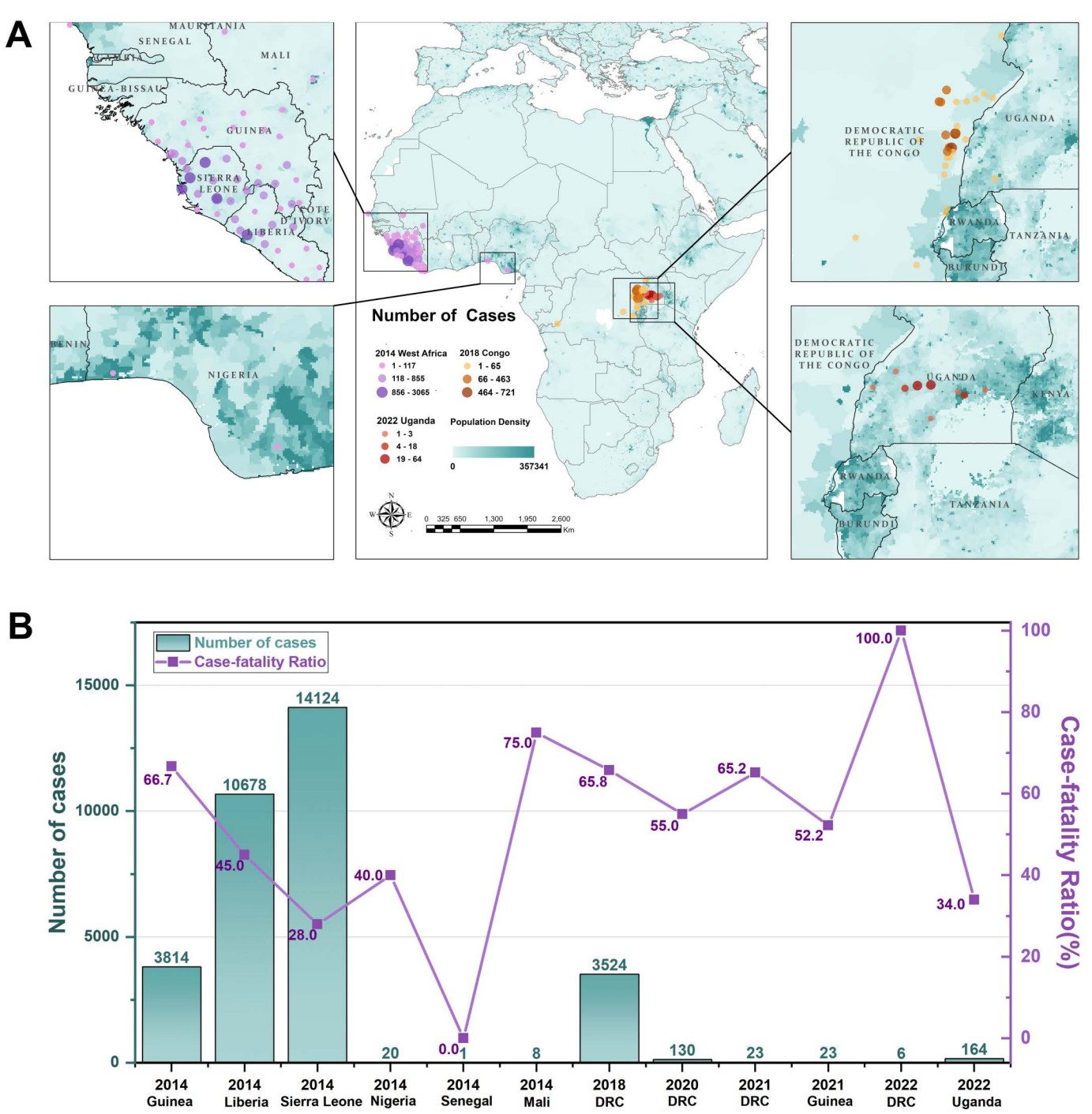

**Fig 1. Study area and spatial-temporal distribution of EVD in Africa since 2014 (A) Spatial distribution of EVD cases, and (B) Temporal variations of EVD cases and fatality ratio in different countries.** The base layers of the map were obtained from the openly available source via the Natural Earth (https://www.naturalearth-data.com/downloads/50m-cultural-vectors/).

**2.2.2.  Environmental data.**  Previous research has demonstrated that the transmission of Ebola is influenced by multiple environmental factors such as climate conditions (e.g., precipitation and temperature) and vegetation abundance (e.g., deforestation and the consumption of forest prey) [24]. In this study, climate data were primarily obtained from WorldClim (https://www.worldclim.org/), a database that offers high-resolution weather and climate data at a global scale.

Surface coverage data were obtained from the Copernicus Climate Change Service (https://climate.copernicus.eu/), which combines climate system observations with scientific research to produce authoritative and quality-assured information on European and global climate. In addition, vegetation information was acquired from the global map data archive established by the Global Map Transfer Program and long-term time series normalized difference vegetation index (NDVI) data were retrieved from NOAA's Advanced Very High Resolution Radiometer (AVHRR) sensor.

**2.2.3.  Human impact data.**  Demographic data were extracted from Gridded Population of the World, version 4 data (GPW.v4) which is based on national censuses and population registers. We also utilized Human Influence Index grids data set which contains information such as population density raster, human land cover raster, and constructed roads. In addition, night time light remote sensing data were also utilized due to its strong correlation with socioeconomic factors, which was obtained from the Operational LineScan System (OLS) of the U.S. Defense Meteorological Satellite Program (DMSP) and the VIIRS Plus DMSP light change dataset.

All the aforementioned data was carefully examined and preprocessed. We cleaned the datasets by removing elements outside normal geographic boundaries and then standardizing raster resolutions to 2.45 km using Python and ArcGIS 10.8. To improve the spatial precision of Ebola-impacted provinces, we utilized the Random Point Generation tool in ArcGIS 10.8 to randomly generate additional points within a 0.3 arc minute rectangular frame around the center of each province, based on the proportion of cases relative to the total epidemic counts. Subsequently, a geospatial database was established in ArcGIS 10.8, integrating epidemic case data, remote sensing imagery, surveyed attributes, and spatial vector boundaries into three distinct feature datasets (Ebola case points, multisource raster, and national boundaries). Finally, topological rules were implemented and quality checks was performed to ensure the integrity and accuracy of database, providing reliable inputs for further in-depth analysis.

**2.2.4.  Variable filter.**  To prevent overfitting and enhance the interpretability of the models, variable filtering was conducted to select significant factors. Based on JackKnife method implemented via Maxent software V3.4.3, and XGboost algorithm in Python, we assessed the significance of each factor [25,26]. In order to solve the strong interrelationships among those variables and improve the accuracy, we utilized Spearman's correlation coefficient in Python to remove the variables exhibiting substantial correlation [27,28]. This process can help identify the more representative and significant factors that were used for subsequent modeling and predicting.

## 2.3.  Methods

Spatial-temporal distribution of EVD in Africa from 2014 to 2023 were visualized by ArcGIS 10.8. In order to accurately estimate the current spatiotemporal distribution of EVD as well as its potential risk, we established a synthesized prediction model by integrating four distinct ecological niche approaches which are Maxent, Bioclim, Domain and GARP models. Table A in S1 Appendix shows the advantages and disadvantages of those four models and they complement each other in characteristics. Specifically, the Bioclim model can provide essential boundary conditions based on climatic extremes [29], serving as a foundational layer to define

species' ecological limits. This can be offset by the Maxent model[30], which incorporates the complex, nonlinear ecological interactions that the Bioclim model may overlook, thus enriching the predictive model with greater ecological depth. On the other hand, simplistic the Domain model[31], can offer valuable preliminary insights especially in data-limited situations, establishing a baseline distribution hypothesis that can be refined. Finally, the iterative optimization processes of the GARP model can better refine the initial predictions by systematically improving accuracy through complex rule handling [32]. Therefore, the integrated model by combining these approaches can not only consolidate basic climatic thresholds and ecological interactions but also increase the precision and adaptability in predicting specie distribution under varying environmental conditions.

The Maxent model, implemented using Maxent software V3.4.3, provides probabilities ranging from 0 to 1 which indicates the likelihood of Ebola presence. The Bioclim and Domain models, executed through DIVA-GIS software, yield larger values for regions more prone to Ebola outbreaks. Meanwhile, the GARP model, realized through DesktopGARP 1.1.3 software, categorizes the study area into suitable and unsuitable regions for Ebola occurrence. The description of those four models can be found in the supplementary materials (S2 Appendix). Our proposed synthesized model is given by

$$\widehat{y}_j^i = f_i\left(X_j', \Theta_i\right)$$

where $i$ represents the type of ecological niche model, $j$ represents the prediction unit, $f$ denotes the model mapping, and $X$ is the feature vector used to describe multidimensional influencing factors. After training the model, input features $X_j'$ are applied to each model for prediction, resulting in predicted values $\widehat{y}_j^i$. By adjusting the model parameters $\Theta_i$, we aimed to minimize the difference between the predicted values $f_i\left(X_j, \Theta_i\right)$ and the actual values $y_j$, which is expressed as:

$$\min_{\Theta_i}\left(y_j - f_i\left(X_j, \Theta_i\right)\right)^2$$

Since the Maxnet, Bioclim, and Domain models yield specific values while the GARP model provides logistic variables, it is necessary to standardize these different results to enable comparison. We reclassified the outputs from the first three models into five uniform intervals ranging from 0 to 1 with each interval at 0.2, to align with the logistic variables from the GARP model. Finally, the predicted values of the four models are weighted and summed to obtain the ultimate ecological niche prediction $\widehat{y}_j$. The weights $W_i$ were derived through the Analytic Hierarchy Process (AHP), comprehensively considering factors such as model accuracy, nonlinear data handling capabilities, complexity, data volume requirements, and generalizability [33]. The summation is conducted as follows:

$$\widehat{y}_j^{inter} = \sum_{i=1}^{4} W_i \cdot \widehat{y}_j^i$$

In our study, the dataset was randomly partitioned into a training set (75% of the data) and a test set (25% of the data) for model validation. Each model was constructed using the training set and subsequently evaluated based on the test set. Based on the predictive accuracy of each model, the final outcome was retrieved by incorporating different weights according to the following standards: (1) the plotted Receiver Operating Characteristic curves (ROC); (2) the calculated Area Under the Curve (AUC) values, a widely used metric for evaluating the classification performance of heavily imbalanced data; (3) normalizing the different suitability

representations of into five levels (1 for Not Suitable, 2 for Less Suitable, 3 for Moderate, 4 for More Suitable, and 5 for Suitable); (4) assigning weights to each model based on the corresponding AUC values, which were then standardized using a grid-based calculation method to obtain the final result. A schematic representation of the entire process was depicted in Fig 2.

We applied the Maxent model and Jackknife testing for repeated experiments (100 iterations) to identify the key environmental variables influencing the prediction of Ebola-suitable habitats. To further explore these variables, we performed the Maxent model and GAM (generalized additive model) (see S2 Appendix) to assess how each environmental factor impacts the predicted risk. For the Maxent model, during the modeling process, all environmental variables were held constant at their mean values, except for one specific variable, which was allowed to vary. The resulting Environmental Marginal Response Curves derived from the Maxent model illustrate the relationship between the predicted probability of suitability and variations in this specific environmental factor. These curves provide insight into how ecological viral adaptability corresponds to specific environmental variables.

For the GAM model, we first generated 1,000 random spatial points within the study area using the Create Random Points tool in ArcGIS. Environmental data for these points, along with the predicted suitability values from the Maxent model, were then extracted. The

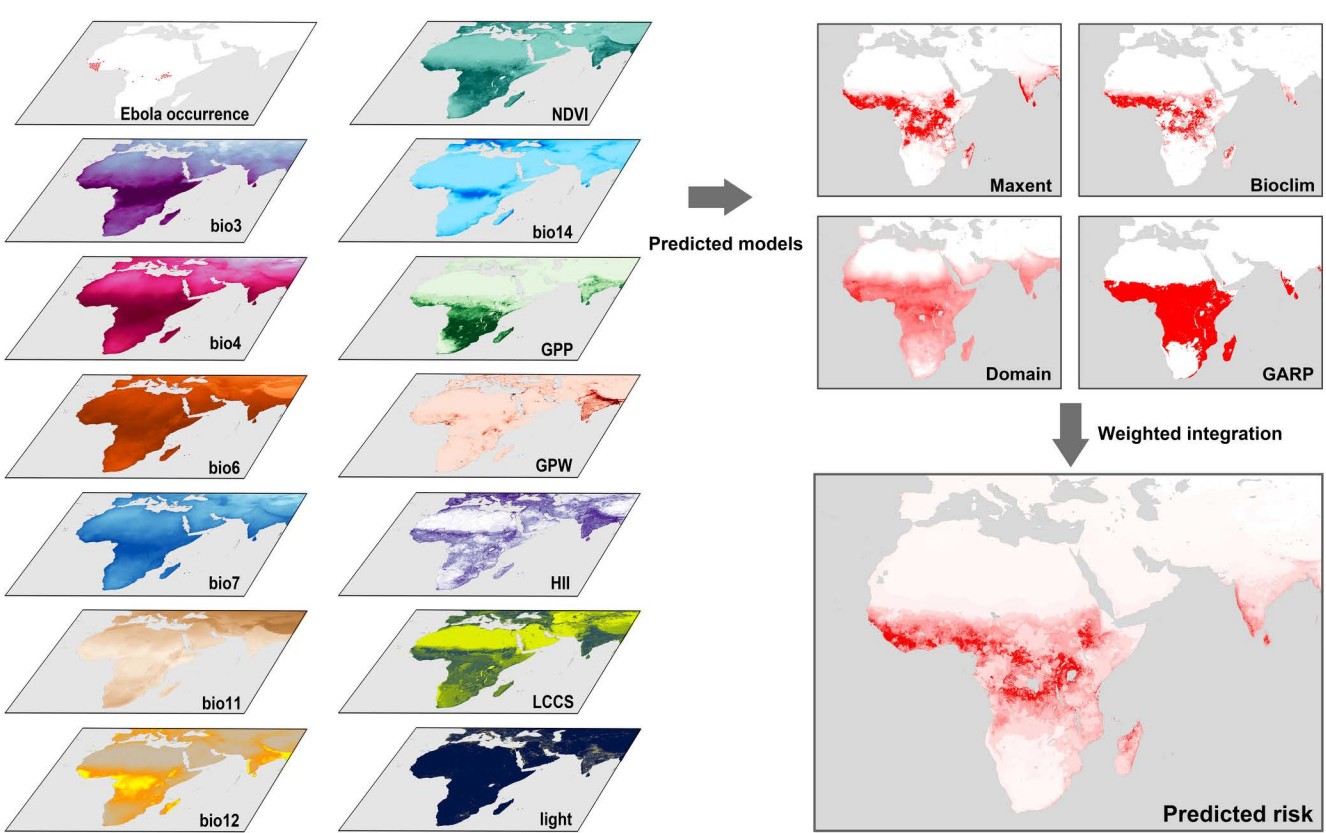

**Fig 2. Predicted ecologically suitable regions of Ebola virus based on Maxent, Bioclim, Domain, and GARP models with 13 independent, influential variables, i.e., bio3, bio4, bio6, bio7, bio11, bio12, bio14, NDVI, GPP, GPW, HII, LCCS, light.** The base layers of the map were obtained from the openly available source via the Natural Earth (https://www.naturalearthdata.com/downloads/50m-cultural-vectors/).

predicted suitability values from the Maxent model served as the response variable in the subsequent GAM analysis, enabling the assessment of the influence of environmental predictors on the modeled suitability.

## 3. Results

### 3.1. Ecological adaptability assessment of Ebola

We ultimately selected 13 variables that are most likely to influence Ebola distribution, namely Isothermality (bio3), Temperature Seasonality (bio4), Min Temperature of Coldest Month (bio6), Temperature Annual Range (bio7), Mean Temperature of Coldest Quarter (bio11), Annual Precipitation (bio12), Precipitation of Driest Month (bio14), Gross Primary Productivity (GPP), Gridded Population of the World (GPW), Human Influence Index (HII), Land Cover Classification System (LCCS), Night Lights (light) and NDVI. Details regarding the units, ranges, standard deviations (STD), means, and data sources of these variables are provided in Table 1. The comprehensive definitions and descriptions of each variable can be referred to the official resource (https://www.worldclim.org/data/bioclim.html).

Each of the four models demonstrate distinctive characteristics for predicting Ebola occurrences across different regions. The reclassified values derived from the first three models are depicted in Fig 3A, 3B and 3C, with darker colors indicating higher adaptability of Ebola. Results of the GARP model in the format of logistic variables are shown in Fig 3D.

The Maxent and Bioclim models show similar predictions for favorable virus habitats, primarily concentrating in West Africa, sub-Saharan nations, along the edge of the Congolese basin, around Lake Victoria, and on the Ethiopian plateau. These ecologically suitable zones are generally situated near the equator. In contrast, the Domain and GARP models utilize more stringent criteria to identify the non-viable areas, and predict a broader range of potential zones distributed across the African continent, spanning from sub-Saharan regions to the Kalahari Desert in the South African plateau.

**Table 1. Description of the selected 13 variables for this study, including their units, ranges, STD, means, and data sources.**

| Variables | Minimum | Maximum | Mean | STD | Source |
|---|---|---|---|---|---|
| Isothermality (bio3) | 9.1 | 100.0 | 34.4 | 18.8 | https://www.worldclim.org/data/worldclim21.html |
| Temperature Seasonality (×100; bio4) | 0 | 2377.6 | 888.5 | 467.8 | |
| Min Temperature of Coldest Month (°C; bio6) | -72.5 | 26.5 | -20.2 | 26.0 | |
| Temperature Annual Range (°C; bio7) | 1 | 72.7 | 34.1 | 12.0 | |
| Mean Temperature of Coldest Quarter (°C; bio11) | -66.4 | 29.3 | -14.3 | 26.7 | |
| Annual Precipitation (mm; bio12) | 0 | 11246.0 | 537.2 | 636.5 | |
| Precipitation of Driest Month(mm; bio14) | 0 | 507.0 | 14.7 | 27.6 | |
| Gross Primary Productivity(GPP) | 0 | 715.2 | 44.2 | 85.7 | https://essd.copernicus.org/articles/14/1063/2022/essd-14-1063-2022-assets.html |
| Gridded Population of the World(GPW) | 0 | 357341.3 | 46.2 | 423.7 | https://sedac.ciesin.columbia.edu/data/set/gpw-v4-population-count-rev11 |
| Human Influence Index(HII) | 0 | 64 | – | – | https://sedac.ciesin.columbia.edu/data/set/wildareas-v2-human-influence-index-geographic |
| Land Cover Classification System(LCCS) | 10 | 220 | – | – | https://maps.elie.ucl.ac.be/CCI/viewer/ |
| Night Lights(light) | 0 | 255.0 | 4.1 | 20.0 | https://eogdata.mines.edu/products/dmsp/ |
| Normalized Difference Vegetation Index(NDVI) | -32768.0 | 9938.0 | 188.9 | 6370.3 | https://www.ncei.noaa.gov/products/climate-data-records/normalized-difference-vegetation-index |

Abbreviation: STD (Standard Deviation), Minimum (The smallest value in the raster), Maximum (The largest value in the raster), Mean (The average value of the raster).

According to the accuracy assessment of ROC curves illustrated by Fig 3E, 3F, 3G and 3H, the AUC values derived from these four prediction models are all relatively high with Maxent (0.99), Bioclim (0.92), Domain (0.93) and GARP (0.85), respectively. In order to synthesize the predictions of the four ecological niche models, we assigned 30%, 30%, 30%, and 10% weights to the four models mentioned, and obtained the integrated danger levels of Ebola outbreaks as well as the potential habitat of EVD in Africa as shown by Fig 4A, while higher levels indicate a greater risk of an Ebola outbreak.

The areas at most risk predominantly lie in the western, central, and eastern regions of Africa, situated south of the Sahara Desert but close to the equator as illustrated by Fig 4B, 4C, 4D and 4E. Specific countries include Guinea, Liberia, Sierra Leone, Nigeria, DRC, Uganda, Ethiopia, among other. Using the Albers Equal Area Conic projection within corresponding coordinate system, the calculated total area occupied by these high-risk regions (areas above level mmoderately high in the map) are approximately 3, 509, 623.11 km², accounting for nearly 11.61% of Africa's entire land area.

By integrating population density grids, we estimated the total population affected in each country located within the high-risk areas depicted in Fig 5. The red bars in Fig 5C represents countries without reported Ebola outbreaks but with a substantial population at risk of infection. The blue bars signify countries with a history of Ebola outbreaks and documented cases. Based on the current predictive results, countries at higher risk yet without a history of Ebola include Ethiopia, Tanzania, Côte d'Ivoire, Ghana, Cameroon, Rwanda, Burundi, and others.

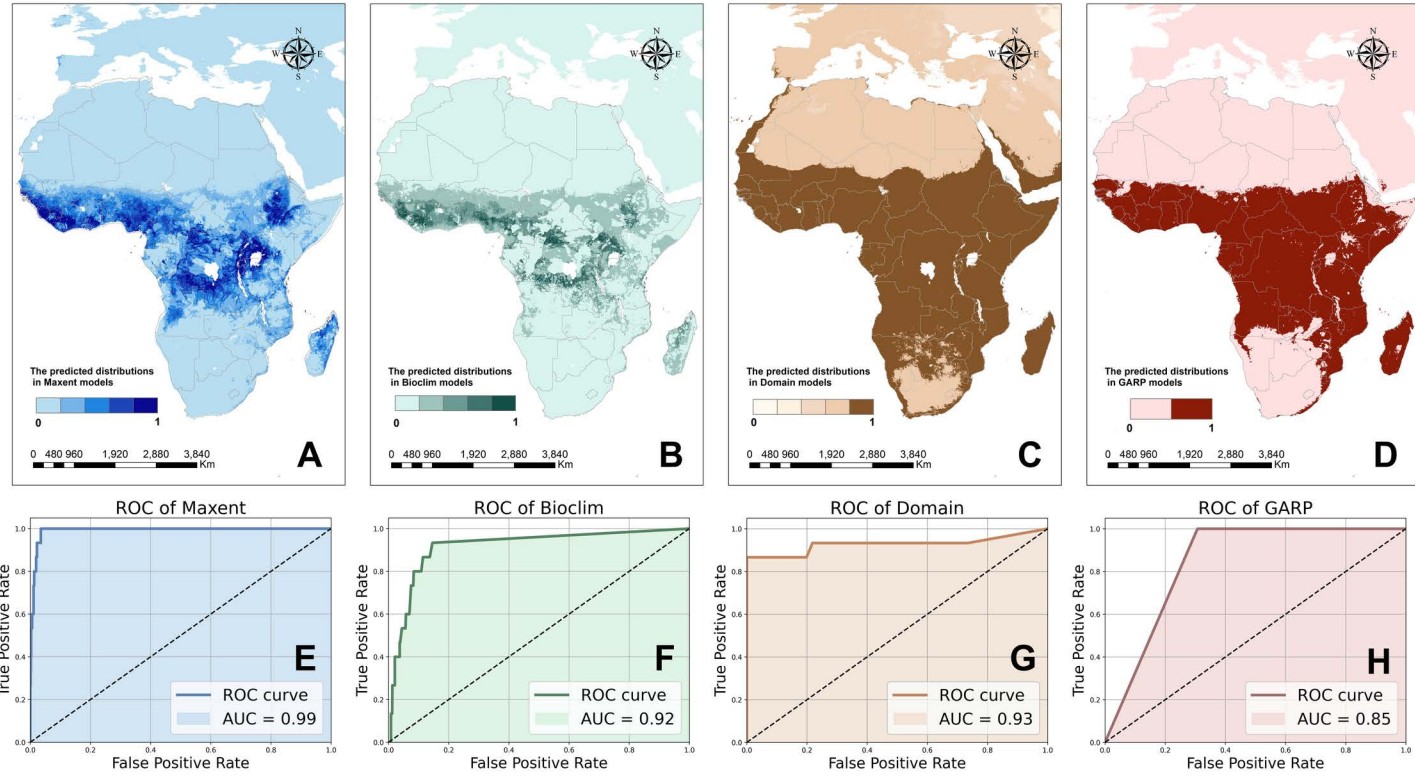

**Fig 3. Potential distribution of Ebola outbreak occurrences based on (A) Maxent(B) Bioclim(C) Domain(D) GARP models plus ROC curves for (E) Maxent(F) Boclim(G) Domain, and (H) GARP.** The base layers of the map were obtained from the openly available source via the Natural Earth (https://www.naturalearthdata.com/downloads/50m-cultural-vectors/).

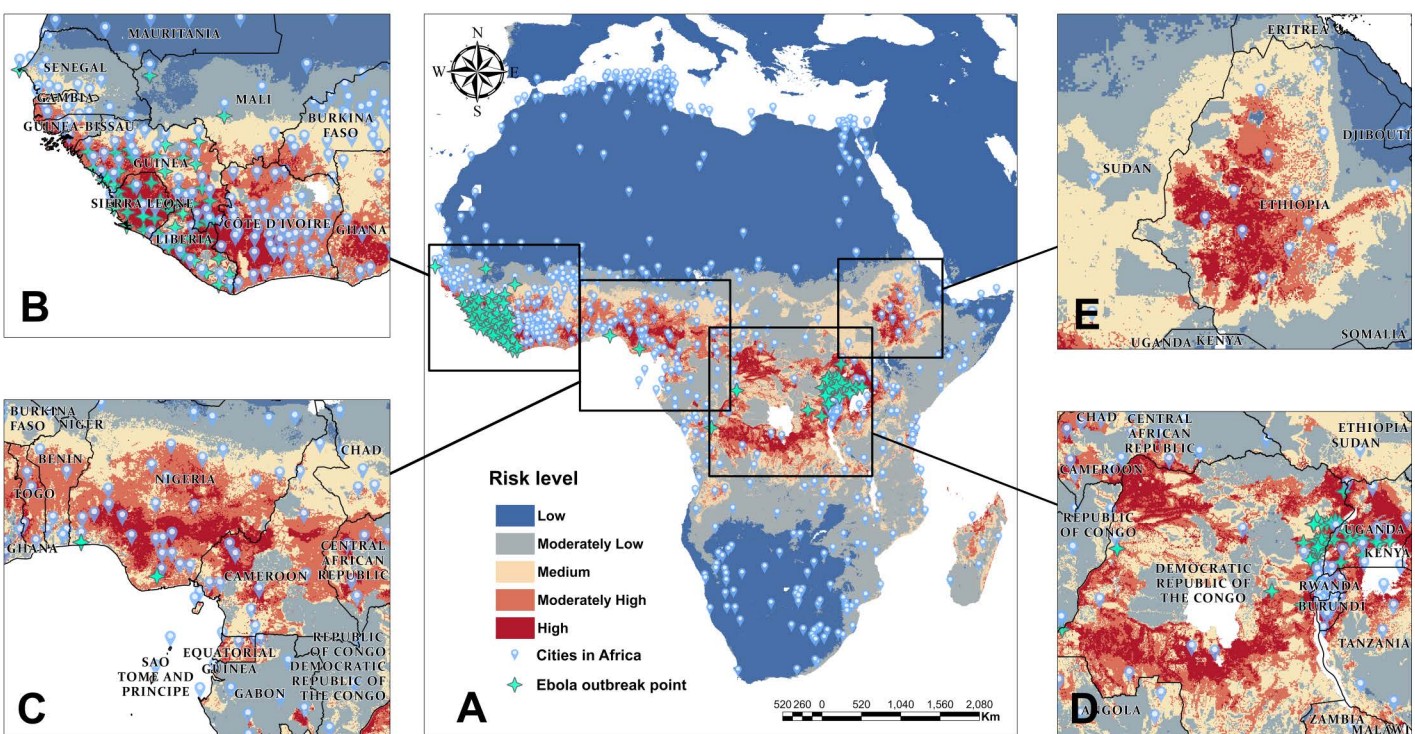

**Fig 4. Map of potentially suitable habitats for Ebola virus in Africa and a closer look at four high-risk regions with (A) Integrated risk levels of Ebola based on results derived from four models(B) Western Africa region(C) Coastal regions of the Gulf of Guinea(D) Central Africa region, and (E) Eastern African region.** The base layers of the map were obtained from the openly available source via the Natural Earth (https://www.naturalearthdata.com/downloads/50m-cultural-vectors/).

Côte d'Ivoire has reported one single Ebola case in 2021 but this did not lead to a further outbreak. Countries that have experienced recent Ebola outbreaks (e.g., Guinea, the DRC, and Liberia), as well as those that had outbreaks in earlier periods (e.g., Gabon, South Africa, and the Sudan), all matched with the predicted Ebola-adapted zones, providing a good validation for the reliability.

### 3.2. Significance analysis of impact factors

Given the superior performance of the Maxent model in accuracy assessment owing to a high AUC value of 0.99, this study conducted an impact factor analysis to examine the crucial environmental variables for predicting Ebola-suitable habitats. Table 2 presents the contribution and permutation importance of 13 environmental variables to the ecological adaptability prediction of EVD, which was derived from Maxent modeling. Averages of 100 experiments reveal that bio4, GPW, bio12, LCCS, and bio6 act in rank as pivotal factors that can affect the suitable niches of Ebola. Those key variables account for a cumulative percentage contribution of 97.8% and permutation importance of 98.5%.

Jackknife testing (Fig 6) based on Maxent software 3.4.3, applied two methods to evaluate the impact of environmental variables on Ebola virus models. The first method involves using only one variable at a time, highlighting the direct impact of variables like bio3, bio4, bio11, bio12, GPP, and GPW on model performance metrics. The second one excludes one variable to assess its unique contribution, revealing that the exclusion of GPW, LCCS, and bio12 significantly reduced model accuracy.

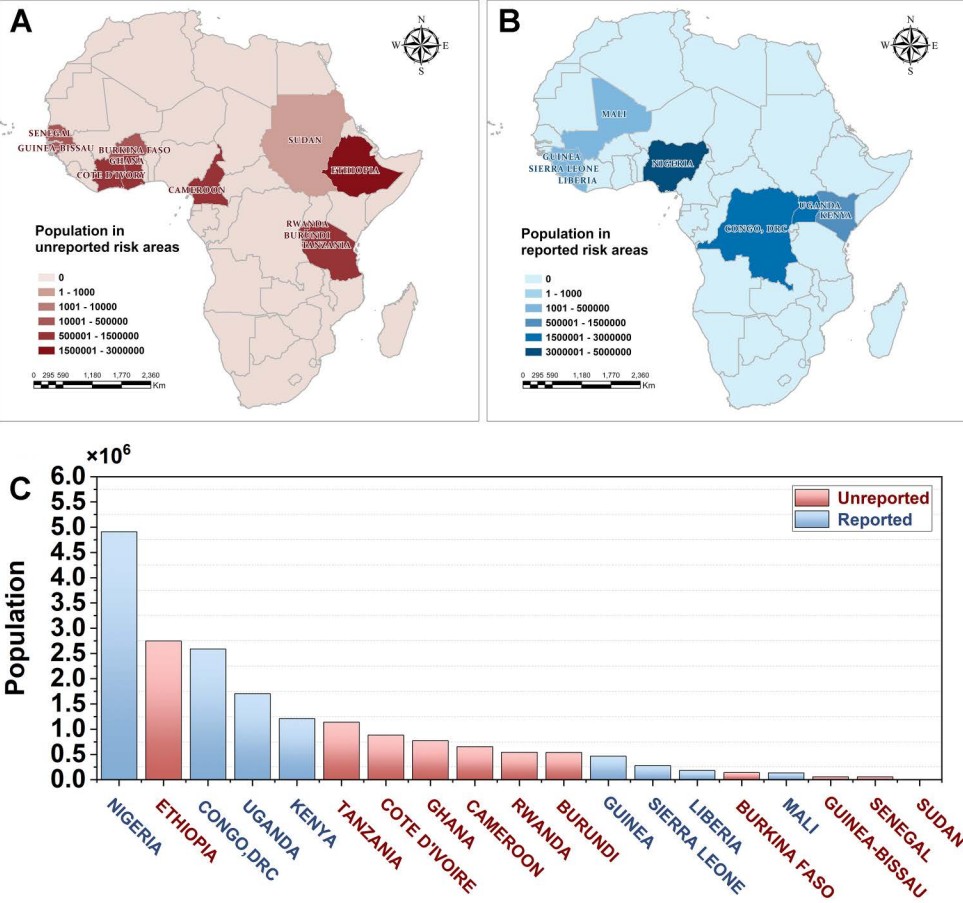

**Fig 5. Populations in Ebola risk areas within each country with (A) Spatial distribution of countries within the Ebola risk zone with reported Ebola outbreak (B) The spatial distribution of countries within the Ebola risk zone without any reported Ebola outbreaks, and (C) Histogram of populations at high risk of Ebola infection.** The base layers of the map were obtained from the openly available source via the Natural Earth (https://www.naturalearthdata.com/downloads/50m-cultural-vectors/).

After combining those findings, it became evident that bio4, bio12 and GPW exert a significant impact on the suitable niches of Ebola.

### 3.3. Response of impact factors

To generate response curves for different variables, we utilized individual variables to construct the Maxent model. Consequently, we obtained the model's predicted probabilities as a function of increasing the values of each environmental factor (Fig 7). The x-axis represents values of the environmental variables, while the y-axis corresponds to the consequent Ebola suitability probability. The line graph demonstrates the dependency of predicted suitability on the selected variable, as well as the dependency arising from the correlation between the chosen variable and other variables. Since HII and LCCS are categorical variables, the results have been illustrated using a standard bar chart.

The GAM model-based impact factors and their response to Ebola's ecological adaptability are shown in Fig 8. Each scatter plot in the figure represents a random point, with the x-axis representing the variable value and the y-axis showing its impact on Ebola's ecological

adaptability. Vertical lines on the x-axis depict data distribution and density, while the light purple background signifies the confidence interval. These lines can represent the fitting of the data points.

**Table 2. Contribution and permutation importance of variables derived through Maxent modeling (average of 100 experiments).**

| Variable | Contribution (%) | Permutation Importance |
|---|---|---|
| bio4(Temperature Seasonality) | 44.89 | 48.19 |
| GPW(Gridded Population of the World) | 33.43 | 32.69 |
| bio12(Annual Precipitation) | 9.76 | 9.09 |
| LCCS(Land Cover Classification System) | 7.21 | 5.67 |
| bio6(Min Temperature of Coldest Month) | 2.47 | 2.82 |
| HII(Human Influence Index) | 1.55 | 0.59 |
| bio3(Isothermality) | 0.52 | 0.44 |
| GPP(Gross Primary Productivity) | 0.14 | 0.43 |
| NDVI(Normalized Difference Vegetation Index) | 0 | 0.06 |
| Light(Night Lights) | 0 | 0 |
| bio7(Temperature Annual Range) | 0 | 0 |
| bio14(Precipitation of Driest Month) | 0 | 0 |
| bio11(Mean Temperature of Coldest Quarter) | 0 | 0 |

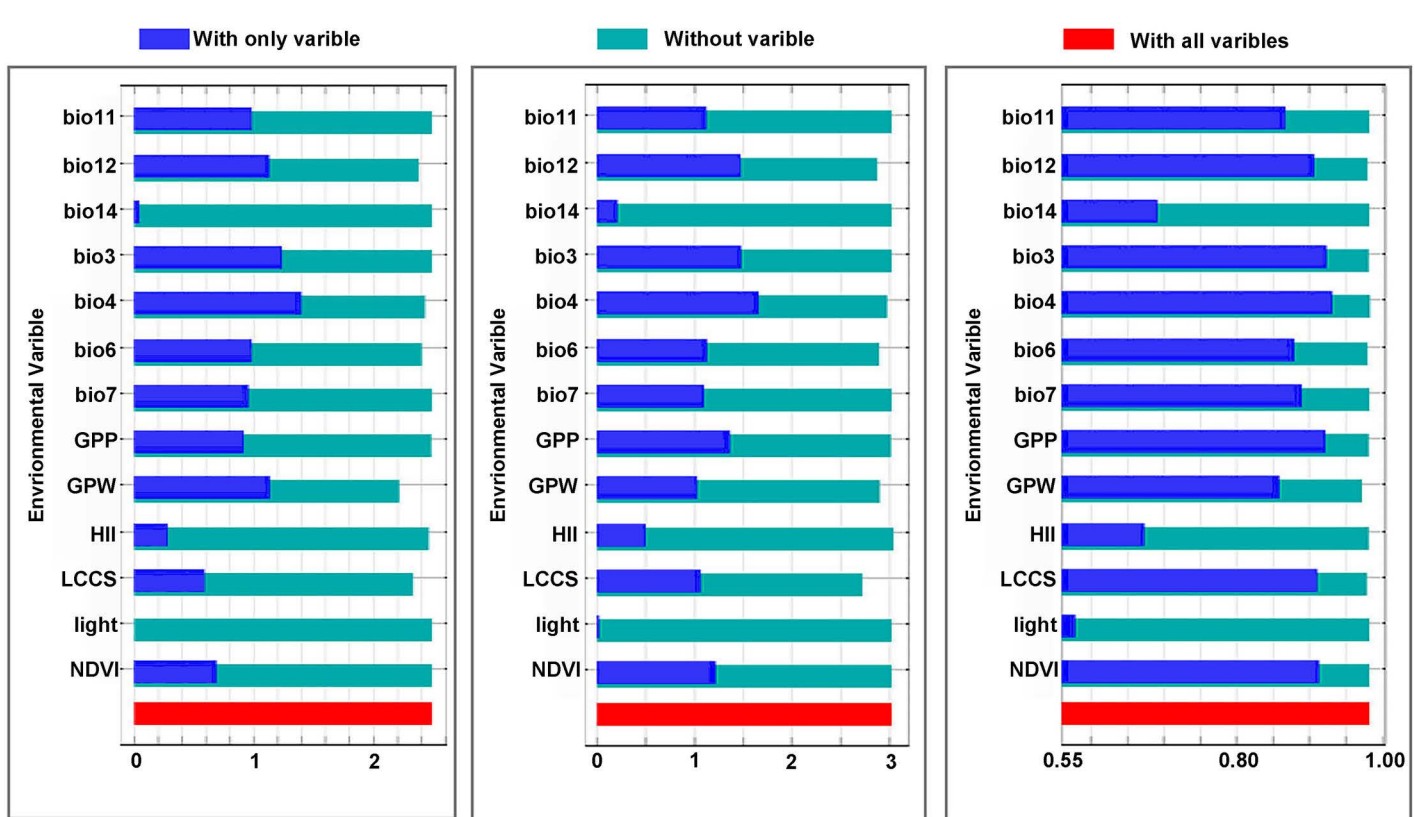

**Fig 6. Jackknife test of the importance of environment variables in Maxent, including Jackknife of regularized training gain, test gain and AUC.**

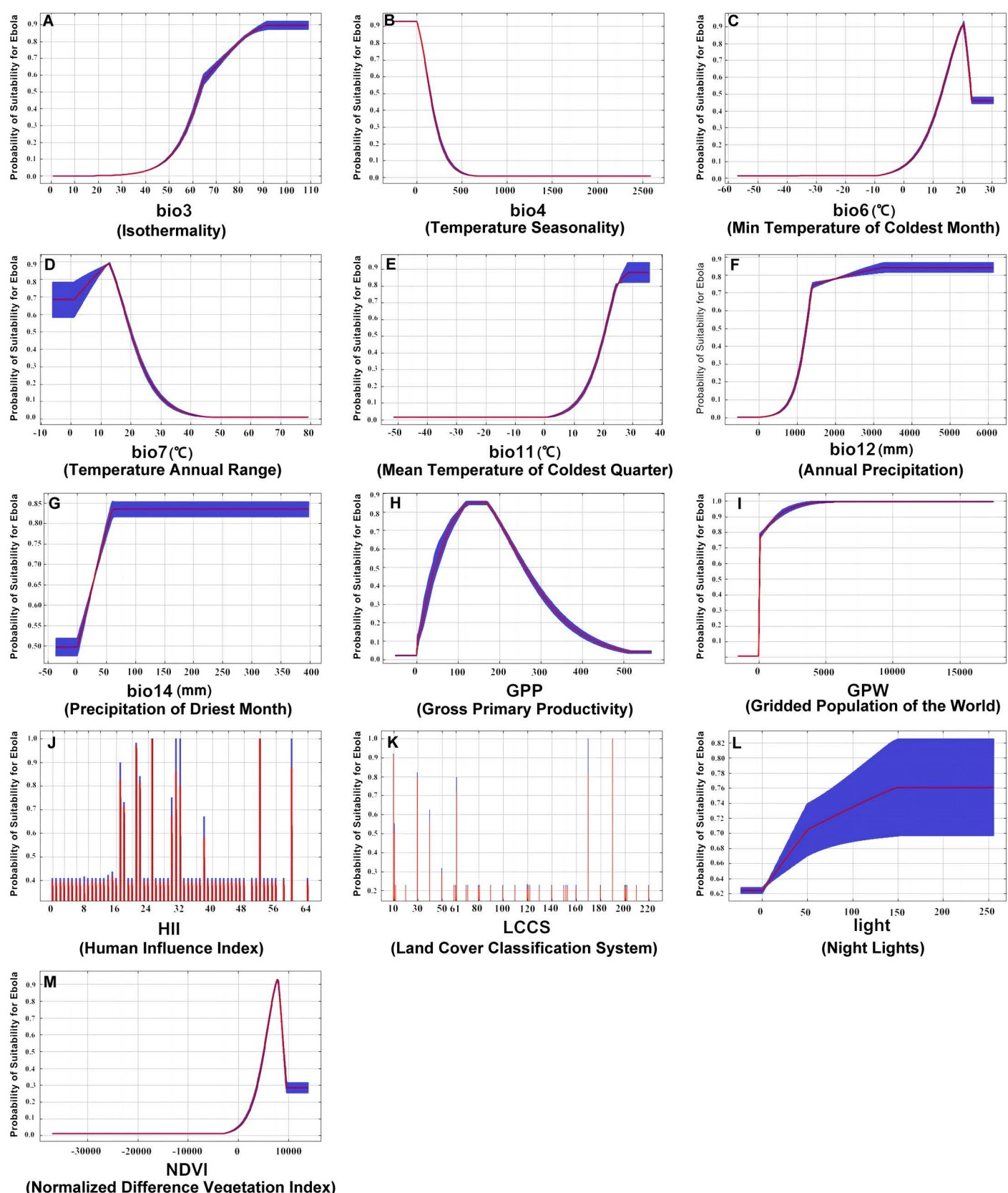

**Fig 7. Response curves of environmental factors and the probability of suitability for Ebola based on Maxent model.** Each curve (A-M) represents model-predicted probabilities associated with bio3, bio4, bio6, bio7, bio11, bio12, bio14, GPP, GPW, HII, LCCS, light, and NDVI. Curves display the mean

response derived from 10 replicate Maxent runs (shown in red) with means and standard deviations (represented in blue, with two shades for categorical variables).

Combining the outcomes yielded by the Maxent model and GAM, we revealed the distinct patterns in Ebola distribution:

(1)  The ecological adaptability of the Ebola virus is inversely related to temperature seasonality (bio4) but positively correlates with higher winter average temperatures and annual precipitation (bio6, bio12).

(2)  Greater population densities enhance the ecological adaptability of the Ebola virus (GPW). However, this adaptability exhibits a bell-shaped curve in relation to economic development, increasing initially before diminishing in more economically advanced regions (HII).

(3)  The ecological adaptability of the Ebola virus shows an initial increase followed by a decline in response to rising vegetation levels, peaking in areas with moderate vegetation coverage.

(4)  The Ebola virus adaptability responds positively to various land covers, with specific codes representing rainfed cropland (10), mosaic cropland (30), broadleaf deciduous forests (60), flooded forests (170), and urban zones (190) (LCCS).

## 4. Discussion

As an interdisciplinary piece of research, this study attempts to investigate suitable ecological niches for Ebola virus as well as its potential responses to different environmental factors. This study provides a practical example of how to apply geostatistical approaches to quantitatively explore the dynamic distribution, variation, and influential determinants of an epidemic, not only demonstrating an opportunity to address certain research gaps in the existing literature, but also presenting some potential to support public health decision-makers in formulating and implementing effective strategies.

Regarding spatiotemporal predictions for Ebola habitats at a macroscopic scale, the forecasted outcomes of the integrated model mainly aggregate to the south of the Sahara Desert and north of Zambia and Angola, covering approximately 16% of the entire African continent. This finding relatively matches with the zoonotic niche map of Ebola virus revealed by Pigott et al. (2014) [14], but in comparison, our high-risk areas are more dispersed and better correspond to the previous outbreak patterns [34]. This may be attributed to the utilization of the latest infection case data, including the 2018 Congolese outbreak and the 2022 Uganda outbreak. Additionally, we incorporated a more comprehensive set of potential influencing factors, which improved identifying the most significant ecological determinants that possibly affect the adaptability of Ebola virus in Africa, thereby further increasing the accuracy of final prediction.

Our research also compiled the demographic statistics within Ebola outbreak-prone regions across different African countries, estimating the total population at risk in these high-risk areas to be 20,504,885. Among this population, 11,464,743 individuals reside in regions with reported Ebola outbreaks, while 9,040,142 individuals live in areas with no reported outbreaks. Notably, Ethiopia, one of the countries with the highest at-risk population, although not previously documented with Ebola cases, has been identified as having the potential risk for an Ebola outbreak in our study, similar to the findings from Redding et al. (2019) [16].

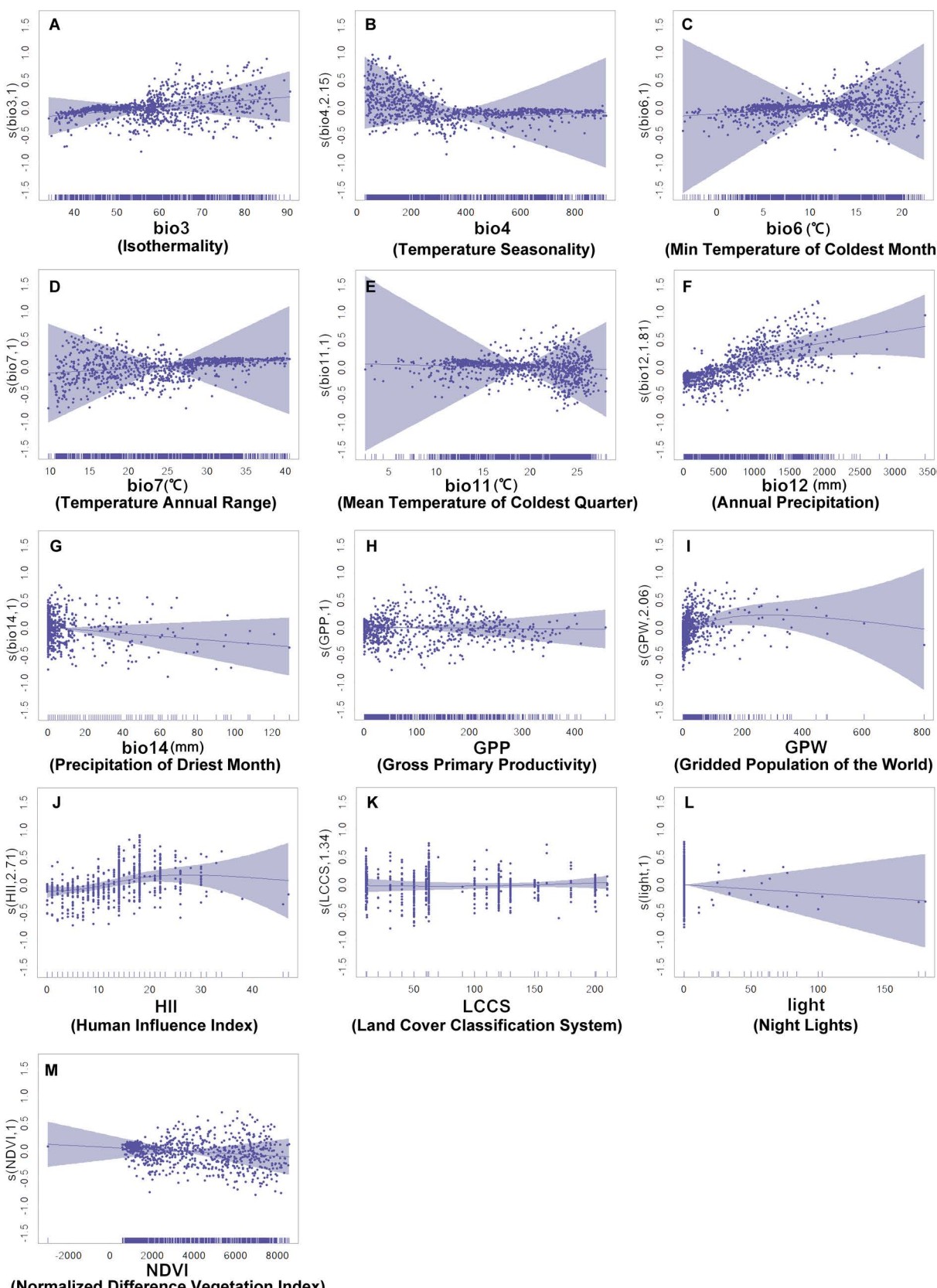

**Fig 8. GAM model-based impact factors and ecological adaptability response of Ebola (A-M), final fitness classes with bio3, bio4, bio6, bio7, bio11, bio12, bio14, GPP, GPW, HII, LCCS, light, and NDVI.**

Both researches highlighted environmental factors in Ethiopia are conducive to a potential Ebola outbreak, warranting serious concern from local governments and global public health institutions. Other countries at high risk of Ebola without previous reports include Tanzania, Cote d'Ivoire, Ghana, Cameroon, and Rwanda, which are urged to promptly implement effective prevention measures against potential Ebola epidemics.

Furthermore, this study scrutinized the responses of various influencing factors and explained the driving effect on Ebola outbreak from a spatiotemporal perspective. Utilizing the Jackknife test and the Maxent model, we identified the key factors closely associated with Ebola's ecological adaptability, notably including bio4, GPW, bio12 and LCCS. In order to further investigate how these influencing factors affect Ebola ecological adaptability, we analyzed the different distribution patterns of the virus derived from the Maxent model and GAM, and provide the following elaborations:

(1) The ecological adaptability of Ebola virus decreases with a rise in temperature seasonality, indicating preference for regions with minimal temperature variation between seasons. This matches closely with the conclusions by Olivero et al. (2017) that the observation of Ebola outbreaks predominantly occurring in low-latitude areas near the equator [35]. Its adaptability also increases with higher winter average temperatures and annual precipitation, similar to the findings on environment temperature and precipitation trends [36].

(2) The ecological adaptability of Ebola virus escalates with greater population density, likely due to the rapid human-to-human virus transmission promoted by frequent interpersonal contacts [37,38]. Additionally, the adaptability increases initially and then decreases, becoming lower in more economically developed regions, which can be explained by the fact that the areas less impacted by anthropogenic activities especially rapid urbanization are almost not susceptible to large-scale outbreaks of Ebola [39], whereas the economically developed regions with better healthcare resources are able to effectively implement quarantine measures to reduce the risk of infection [40].

(3) The ecological adaptability of Ebola virus follows a pattern of growth to decline as vegetation increases, peaking within regions of moderate vegetation coverage. For equatorial area with more vegetation abundance, the reduction in vegetation possibly accelerates the risk of Ebola virus transmission. Other studies have also indicated that deforestation substantially elevates human interaction with infected wildlife, thereby favoring viral spread [41]. This perspective effectively corroborates our findings.

(4) The ecological adaptability of Ebola virus responds positively to land covers with values of 10, 30, 60, 170, and 190 respectively representing rainfed cropland, mosaic cropland, broadleaf deciduous forests, flooded forests and urban zones. This is consistent with the findings of Redding et al. (2019) who claimed that the land-use/land-cover determines distributions across the different reservoir host species [16]. Nevertheless, we took a step further by incorporating a quantitative analysis, particularly focusing on the nuances within land cover types.

Nonetheless, several limitations should be also taken into consideration. Firstly, the accessed Ebola case data used in this study were limited to the provincial-level records without precise geographical coordinates (e.g., longitude and latitude), which hindered the more accurate spatial analysis. Secondly, bats are known as critical hosts of Ebola virus, our evaluation did not take bats into consideration due to the difficulty in obtaining such information. This may lead to some incomplete understanding of how those host factors drive the risk of Ebola outbreaks. Thirdly, from a model perspective, our synthesized model focuses on the static ecological adaptability and does not incorporate spatial interaction dynamics, thereby incapable

to simulate the mechanisms of Ebola virus spread across regions. The proposed framework also lacks a hierarchical multi-scale analytical design, weak in disentangling complex interactions at varying spatial and temporal scales. In addition, utilizing multi-source raster datasets at varying resolutions in one model involved normalization and resampling, and these steps may introduce uncertainties, such as the loss of fine-scale details or interpolation-induced artifacts, which could slightly affect the accuracy of subsequent analyses. Therefore, in further research efforts need to be devoted in optimizing the model function and enlarging the dataset, to provide more insights into the spread route of Ebola disease once spillover has occurred.

## 5. Conclusions

This study integrated different ecological niche models to effectively predict the spatial distribution of suitable habitats for the Ebola virus. Our findings indicate that those suitable habitats are primarily concentrated in regions near the equator, south of the Sahara Desert. Additionally, our research delved into the response patterns of key influencing factors related to Ebola, elucidating specific associations between Ebola outbreaks with variables such as climate, socio-economic conditions, vegetation, and land cover types. These insights offer valuable directions for future research in the field of epidemic prevention and control.

## Supporting information

**S1 Appendix. Supplemental table and figures** . **Fig A.** Multiple data sources, including Ebola epidemic case data, surveyed attribute data, remote sensing data, and spatial vector boundaries, and establishment of a geodatabase. The base layers of the map were obtained from the openly available source via the Natural Earth (https://www.naturalearthdata.com/downloads/50m-cultural-vectors/). **Fig B.** Heat map of spearman coefficients for each variable illustrating the correlations among variables. **Table A.** The advantages and disadvantages of Maxent, Bioclim, Domain, and GARP models.
(DOCX)

**S2 Appendix**. **Characteristics and introduction of the models used in this study** . **Table A.** The format requirements of the input data for models used in this study.
(DOCX)

## Acknowledgments

The project team would like to thank all the participants who consented to participate in the study. We thank our colleagues for their careful reading and editing of this manuscript. The corresponding author had full access to all the data in the study and had final responsibility for the decision to submit for publication.

## Author contributions

**Conceptualization:** Li Shen, Jiawei Song, Yibo Zhou, Zhongjun Shao, Rui Li, Kun Liu.

**Methodology:** Li Shen, Jiawei Song, Yibo Zhou, Xiaojie Yuan, Samuel Seery, Ting Fu, Xihao Liu, Yihong Liu.

**Supervision:** Xihao Liu, Yihong Liu, Zhongjun Shao, Rui Li, Kun Liu.

**Writing – original draft:** Li Shen, Jiawei Song, Yibo Zhou.

**Writing – review & editing:** Xiaojie Yuan, Samuel Seery, Ting Fu, Zhongjun Shao, Rui Li, Kun Liu.

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
