## [Decision Letter · Decision Letter 0]

6 Aug 2024

Dear Mr Li,

Thank you very much for submitting your manuscript "Assessing the ecological resilience of Ebola virus in Africa and potential influencing factors based on a synthesized model" for consideration at PLOS Neglected Tropical Diseases. As with all papers reviewed by the journal, your manuscript was reviewed by members of the editorial board and by several independent reviewers. In light of the reviews (below this email), we would like to invite the resubmission of a significantly-revised version that takes into account the reviewers' comments.

We cannot make any decision about publication until we have seen the revised manuscript and your response to the reviewers' comments. Your revised manuscript is also likely to be sent to reviewers for further evaluation.

Sincerely,

Mabel Carabali, M.D., M.Sc., Ph.D.,

Section Editor

Mabel Carabali

Section Editor

Reviewer's Responses to Questions

**Key Review Criteria Required for Acceptance?**

**Methods**

-Are the objectives of the study clearly articulated with a clear testable hypothesis stated?

-Is the study design appropriate to address the stated objectives?

-Is the population clearly described and appropriate for the hypothesis being tested?

-Is the sample size sufficient to ensure adequate power to address the hypothesis being tested?

-Were correct statistical analysis used to support conclusions?

-Are there concerns about ethical or regulatory requirements being met?

Reviewer #1: None

Reviewer #2: The objectives are clearly stated, however the methodology and data are not clearly explained.

Line 108: The final number of cases and deaths according to WHO was 3481 and 2291 respectively. Please update and provide a citation

Line 112: Please provide a citation to support this statement: I find it very surprisingly that there would be an established epidemiological link between the 2014 West African outbreak and the 2022 Uganda outbreak given that the former was the Zaire subtype whilst the latter was the Sudan subtype.

Line 120: Please provide a link and citation to the actual dataset used, not just the platform. What variables did you extract: case numbers, deaths, or death? What temporal resolution (if any) did you use?

Line 126: Again, please provide a link to the exact data source, not just the data platform

Line 136: This is not the correct reference

Line 137: Please link to the exact database and provide more details on the variable used, their meanings and the resolution of the data. Please do the same for the CCCS, NDVI and AVHRR data. Did you consider any variables related to bat habitats, given they are a reservoir for Ebola?

Section 2.2.3: Please provide citations and links for GPW, HII, DMSP and VIIRS data, as well as details of the variables extracted and their resolution. How was data cleaning done? What software did you use? What variables had to be cleaned? How did you establish the geospatial database?

Section 2.2.4: Please provide references for the methods and tools used? Did you use pre-existing software packages or write your own? What language did you use? What what the correlation threshold used to remove variables? How sensitive are the results to the threshold used? The 13 variables listed here that are likely to influence Ebola transmission really should be part of the results, not methods. But it would be helpful to understand the full list of variables and what they mean. For example, how are temperature seasonality and isothermality defined? How is the coldest quarter defined, given that Sub-Saharan Africa cannot be divided into four distinct seasons?

Section 2.3: Details on the model are way too sparse. I understand there might not be space to provide a huge amount of detail in the main text, but I think a brief description of the models, their differences and why one would want to combine them rather than perform a model selection is necessary. The details in the appendix are also much too sparse. A general outline of the models is provided, but is too generic. How do you use the models with your data specifically?

How is the model trained? What formula do you use to calculate the weights?

**Results**

-Does the analysis presented match the analysis plan?

-Are the results clearly and completely presented?

-Are the figures (Tables, Images) of sufficient quality for clarity?

Reviewer #1: (No Response)

Reviewer #2: It is very hard to assess the results given that the figures provided do not generally correspond with the figures referred to in the text.

Section 3.1: The first paragraph on reclassification should really be part of the methods section, and more details on how this reclassification is done are needed. Figure 2 does not have subfigures labelled a, b and c.

Figures 3a, b and c are never referred to - maybe rethink whether they need to be included?

How were the weights chosen calculated?

Please label your figures carefully - I'm guessing figures s2 and s3 should actually be figures 5 and 6? Are the countries in red bars actually at risk of Ebola outbreaks? Have you taken into account bat habitats and whether bats live there?

Section 3.2: How was the impact factor analysis done? More detail on how the variable interact with risk is needed. The Jackknife results are in figure 4, not figure 3. Reference 20 is probably not correct.

Section 3.3: I think figure S4 should be figure S1 and figure 4 should be figure 7? Please describe the relationship between the variables and outbreak risk? Are they significant? It is interesting that risk is high for low and high LCCS but not intermediate values - do you have any insight into why? Also, please replace bio3, bio4, bio6, bio7, bio11, bio12, GPP, GPW, HII, LCCS, light and NDVI with more descriptive names. What do the range of these variables represent? What are their units?

**Conclusions**

-Are the conclusions supported by the data presented?

-Are the limitations of analysis clearly described?

-Do the authors discuss how these data can be helpful to advance our understanding of the topic under study?

-Is public health relevance addressed?

Reviewer #1: (No Response)

Reviewer #2: The papers by Pigott and Redding should really be discussed in the introduction.

Patterns 1-4 should really be described in the results, not in the discussion. Their significance and relevance can be assessed in the discussion.

**Editorial and Data Presentation Modifications?**

Reviewer #1: (No Response)

Reviewer #2: (No Response)

**Summary and General Comments**

Reviewer #1: Shen L et al. have assessed the ecological resilience of Ebola virus in Africa as well as its potential influencing factors based on developing a synthesized model. It is of much scientific significance to reveal the spatiotemporal distribution pattern of Ebola and how it responds to different environmental factors. I’m very glad to review the paper in great depth because the study is very interesting. The submission is worth of publication, and following are some minor comments:

1. It is advisable to simplify the description of EVD in the first paragraph (Line 59-63) to make it more succinct.

2. The literature review of different models in the application of EVD research needs to be shortened and succinct (Line 70-86).

3. Some of the figures need to be improved by using high resolution. Especially when it comes to zooming in those figures, a part of information looks really blur.

4. References need to be given to the four ecological niche models in Line 180.

5. Since the full names of different variables have been mentioned in Line 169-174, there is no need to repeat the full name for some variables in Line 279-281.

6. It is suitable to add a Data availability statement at the end of the research part to provide support for researchers who have an interest on this study.

7. One or more pertinent/related scholarly references could be cited in the current reference list.

8. Please double check the grammar mistake and carefully revise the English writing.

Reviewer #2: This paper presents a model combining 4 different ecological models in order to assess and predict the Ebola virus' ecological niche in Sub-Saharan Africa. I personally find this topic very interesting. As the authors point out, a lot of mathematical models of Ebola focus on transmission dynamics predicated on a spill over event happening, however are unsuitable to model the actual spill over event itself. Given land use, climate and population changes over the next few decades, it will be important to understand the spatiotemporal risk of Ebola epidemics in order to prepare and respond to them.

However, whilst I am interested in and consider this topic to be important, this paper is not currently ready for publication. In addition to the comments above, there are several issues that I would like the authors to address:

1. The introduction fails to motivate the study and demonstrate its novelty. The authors discuss the current state of Ebola modelling by solely discussing transmission modelling (and citing a very limited subset of methods and papers at that) - however, they do not really mention previous ecological models in this section. Based on this, the reader may think that this is the first time an ecological niche modelling (ENM) approach has been used in the context of Ebola outbreaks. In the discussion section, the authors do refer to two other ecological models which agree with their findings. However, these models really need to be discussed in the introduction section. What has been done before? What approaches have been used? What were these studies' limitations and how does your model specifically advance our understanding of the Ebola virus' ecological niche. In addition to Redding et al and Piggott et al, other papers to consider include: Judson et al (PLOS Pathogens 2016), Lee-Cruz et al (PLOS NTDs 2021), Buceta et al (PLOS One 2017), to name just a few.

2. More care needs to be taken with citations. There are some statements made in the paper that really need to be backed up with citations. Other papers are miscited. For example, a paper by Merler at el is cited to show that vegetation and climate both impact Ebola transmission; however this paper is an agent-based model on Ebola transmission that assesses the impact of non-pharmaceutical interventions and makes no mention of climate. Similarly, Guo et al is cited to suggest that specific variables are important for Ebola transmission, but is really about the spatial distribution of a plant species.

3. Much more care needs to be taken with figures. The submitted article contained two lists of figure legends: one for figures 1-4 and S1-S3, and one for figures 1-7 and S1. However, none of these seem to match up with the figures accompanying the paper (labelled 1 to 7 and S1). For example, the first list of figure legends says that figure 3 shows the results of a Jackknife test, while the second set of figure legends says that figure 6 shows the Jackknife test. Having looked at the figures, it is actually figure 4 that shows the results of the Jackknife test. It is virtually impossible for the reader to understand whether the results support the conclusions when some figures are never referred to, some figures that are referred to are not shown, and figures are generally mislabelled.

4. More care also needs to be taken with the paper organisation: the methods section contains some results, the results section contains some methods, and results are also presented in the discussion.

I would be amenable to reading a revised version of this paper as I believe it is an important topic. However, it is not ready for publication in its current form.

PLOS authors have the option to publish the peer review history of their article (what does this mean? ). If published, this will include your full peer review and any attached files.

**Do you want your identity to be public for this peer review?** For information about this choice, including consent withdrawal, please see our Privacy Policy .

Reviewer #1: No

Reviewer #2: No
---

## [Decision Letter · Decision Letter 1]

13 Nov 2024

PNTD-D-24-00256R1Assessing the ecological resilience of Ebola virus in Africa and potential influencing factors based on a synthesized modelPLOS Neglected Tropical Diseases Dear Dr. Li, Thank you for submitting your manuscript to PLOS Neglected Tropical Diseases. After careful consideration, we feel that it has merit but does not fully meet PLOS Neglected Tropical Diseases's publication criteria as it currently stands. Therefore, we invite you to submit a revised version of the manuscript that addresses the points raised during the review process. Please submit your revised manuscript within 60 days Dec 13 2024 11:59PM. If you will need more time than this to complete your revisions, please reply to this message or contact the journal office at plosntds@plos.org. Please include the following items when submitting your revised manuscript:* A rebuttal letter that responds to each point raised by the editor and reviewer(s). You should upload this letter as a separate file labeled 'Response to Reviewers '. This file does not need to include responses to any formatting updates and technical items listed in the 'Journal Requirements' section below.* A marked-up copy of your manuscript that highlights changes made to the original version. You should upload this as a separate file labeled 'Revised Manuscript with Track Changes '.* An unmarked version of your revised paper without tracked changes. You should upload this as a separate file labeled 'Manuscript '. If you would like to make changes to your financial disclosure, competing interests statement, or data availability statement, please make these updates within the submission form at the time of resubmission. Guidelines for resubmitting your figure files are available below the reviewer comments at the end of this letter. We look forward to receiving your revised manuscript. Kind regards, Mabel Carabali, M.D., M.Sc., Ph.D.,Section EditorPLOS Neglected Tropical Diseases Mabel CarabaliSection EditorPLOS Neglected Tropical Diseases

Shaden Kamhawi

co-Editor-in-Chief

Paul Brindley

co-Editor-in-Chief

 **Additional Editor Comments (if provided):** Thank you for submitting your manuscript. Although it has been read with great interest, there are several aspects related to the methodology that are deserving special attention. For instance, aspects of the data set and the splitting data for training and validation are not clear, as it is the case with mismatching results and analysis.  **Journal Requirements:****Reviewers' Comments:** Reviewer's Responses to Questions

**Key Review Criteria Required for Acceptance?**

**Methods**

-Are the objectives of the study clearly articulated with a clear testable hypothesis stated?

-Is the study design appropriate to address the stated objectives?

-Is the population clearly described and appropriate for the hypothesis being tested?

-Is the sample size sufficient to ensure adequate power to address the hypothesis being tested?

-Were correct statistical analysis used to support conclusions?

-Are there concerns about ethical or regulatory requirements being met?

Reviewer #1: Yes

Reviewer #2: The objectives of the study are clearly explained. I'd like to thank the authors for updating the methods sections to provide further details, but I would still like a bit of clarification on a couple of points:

Line 120: the data set referenced here doesn't list Ebola cases, rather it lists the health facilities in DRC. Please can you check this and provide the correct data reference.

Line 124: I don't think the description of HDX really adds anything here as you don't use data from over 250 countries and the resolution varies by data set.

Line 160: I'm unclear about the purpose and method behind generating extra points via random sampling.

Line 209: You mention training the model. This would imply a training data set and a testing data set. Please can you provide further details of this?

Figure 1: I think more thought needs to be put into this figure and what you want it to show. The time scale is kind of confusing, as well as the way that total case numbers are assigned to the month of outbreak in a country. It also gives the impression that Guinea, Liberia and Sierra Leone are separate outbreaks whereas they are all part of the same outbreak. Two separate outbreaks in DRC have been conflated (one was declared in May 2018, one in August 2018). I'm not sure that CFR (should be ratio rather than rate!) is helpful or relevant to the study. The CFR for the single case in Senegal (which is again, really part of the West Africa outbreak) should either be 0 or 1, not 0.4!

General: Thank you for providing more information on the data and the models (both in the text and Appendix B). However, for anyone hoping to repeat your analysis, I think it is still hard to understand what the models need in terms of data and what the model parameters are. I understand that the models that previously been published in Ecology journals (thank you for adding references), but they will likely be new to readers of PLOS NTDs. Given the data described so far, it is unclear to me how the input data requirements for the Maxent model (with influential factors x and species occurrence y) corresponds to the points A and B in p-dimensional space for the bioclim model? Do the models require the same or different data processing steps. How many parameters does each model have?

**Results**

-Does the analysis presented match the analysis plan?

-Are the results clearly and completely presented?

-Are the figures (Tables, Images) of sufficient quality for clarity?

Reviewer #1: Yes

Reviewer #2: The results generally match the analysis plan. However, given that in the introduction, one of the main motivations for combining four distinct models was that relying on a single specific model may cause a decrease in accuracy due to inherent limitations, I would have expected to see some kind of analysis showing the benefit (or lack thereof) or synthesizing models as oppose to relying on one.

Line 302: What are the 100 experiments and how are they conducted? Did I miss something in the methods section?

Line 327: How where the 1000 points generated?

I also think the image quality needs to be improved (see Editorial and Data presentation modifications?)

**Conclusions**

-Are the conclusions supported by the data presented?

-Are the limitations of analysis clearly described?

-Do the authors discuss how these data can be helpful to advance our understanding of the topic under study?

-Is public health relevance addressed?

Reviewer #1: (No Response)

Reviewer #2: In the abstract and conclusions, the authors mention that their findings suggest new way to respond to small outbreaks, but nothing about this and public health is really discussed.

The limitations focus on the limitations of data availability and don't really think about limitations of the models, what they can or can't do and what errors or biases different stages of the data processing might introduce.

I also think the study is conflating two 'ecological' factors: one is where the zoonotic spillovers might occur, and one is where the disease might spread when the spillover has already occurred. I think this issue needs a more nuanced response.

**Editorial and Data Presentation Modifications?**

Reviewer #1: Accept

Reviewer #2: Many thanks to the authors for updating the figure numbers. However, the figure resolution is pretty low and it is hard to make out the names of the countries in figures 1, 4, and 5, and the scales in figures 7 and 8. These scales should also have units and y axis labels (at least for the leftmost figures.)

I'm grateful to the authors that they have included a key for the variable names in table 1. However, if would also be helpful to have the proper variable names in the figures and table 2 so that the reader doesn't keep have to referring back to table 1 in order to understand the results.

For table 1, a couple of the data sources are missing, and I would like some clarity of the meaning and relevance of the means, maximums, minimums, etc. The area chosen for the study is Africa, but these values can't belong to Africa. Please can you check and amend these?

**Summary and General Comments**

Reviewer #1: None

Reviewer #2: I think there is still some confusion over what the aim of this paper is and what it does that is not already in the literature. The authors claim that other studies do not take into account spatiotemporal trends in EBOV adaptability, whereas I believe some of them do. I think there is a bit of confusion over modelling zoonotic spillover, versus modelling human to human transmission that needs to be more carefully though about. The authors claim that synthesizing 4 ecological models provide a better understanding of Ebola's niche, but don't really demonstrate it. I suggest that they think a little bit more about where they want to position this paper within the literature and focus on achieving those aims first and foremost.

PLOS authors have the option to publish the peer review history of their article (what does this mean? ). If published, this will include your full peer review and any attached files.

**Do you want your identity to be public for this peer review?** For information about this choice, including consent withdrawal, please see our Privacy Policy .

Reviewer #1: No

Reviewer #2: No

---

## [Editor Report · Decision Letter 2]

15 Jan 2025

Dear Mr Li,

We are pleased to inform you that your manuscript 'Assessing the ecological resilience of Ebola virus in Africa and potential influencing factors based on a synthesized model' has been provisionally accepted for publication in PLOS Neglected Tropical Diseases.

Best regards,

Mabel Carabali, M.D., M.Sc., Ph.D.,

Section Editor

Shaden Kamhawi

co-Editor-in-Chief

Paul Brindley

co-Editor-in-Chief

---

## [Editor Report · Acceptance letter]

Dear Mr Li,

We are delighted to inform you that your manuscript, "Assessing the ecological resilience of Ebola virus in Africa and potential influencing factors based on a synthesized model," has been formally accepted for publication in PLOS Neglected Tropical Diseases.

Best regards,

Shaden Kamhawi

co-Editor-in-Chief

Paul Brindley

co-Editor-in-Chief
